# Double High-Level Ozone and PM$_{2.5}$ Co-Pollution Episodes in Shanghai, China: Pollution Characteristics and Significant Role of Daytime HONO

**Kejing Yang [1], Lingdong Kong [1,2,*], Songying Tong [1], Jiandong Shen [3], Lu Chen [1], Shengyan Jin [1], Chao Wang [1], Fei Sha [4] and Lin Wang [1]**

[1] Shanghai Key Laboratory of Atmospheric Particle Pollution and Prevention, Department of Environmental Science & Engineering, Jiangwan Campus, Fudan University, Shanghai 200438, China; 17210740017@fudan.edu.cn (K.Y.); ldkong@fudan.edu.cn (L.K.); 17210740012@fudan.edu.cn (S.T.); 18210740037@fudan.edu.cn (L.C.); 19210740019@fudan.edu.cn (S.J.); 19210740057@fudan.edu.cn (C.W.); lin_wang@fudan.edu.cn (L.W.)

[2] Institute of Eco-Chongming, East China Normal University, No. 3663 Northern Zhongshan Road, Shanghai 200062, China

[3] Hangzhou Environmental Monitoring Center, Hangzhou 310007, China; 052047010@fudan.edu.cn

[4] Pudong New Area Environmental Monitoring Station, No. 51 Lingshan Road, Shanghai 200135, China; pdsf@sina.com

\* Correspondence: ldkong@fudan.edu.cn

**Abstract:** In recent years, high fine particulate (PM$_{2.5}$) pollution episodes with high ozone (O$_3$) levels have been observed in Shanghai from time to time. However, their occurrence and characteristics remain poorly understood. Meanwhile, as a major precursor of tropospheric hydroxyl radical (OH) that initiates the formation of hydroperoxyl and organic peroxy radicals, HONO would inevitably affect the formation of O$_3$, but its role in the formation of O$_3$ during the double high-level PM$_{2.5}$ and O$_3$ pollution episodes remains unclear. In this study, the characteristics of the double high pollution episodes and the role of HONO in O$_3$ formation in these episodes were investigated based on field observation in urban Shanghai from 2014 to 2016. Results showed that high PM$_{2.5}$ pollution and high O$_3$ pollution could occur simultaneously. The cases with data of double high O$_3$ and PM$_{2.5}$ concentrations accounted for about 1.0% of the whole sampling period. During the double high pollution episodes, there still existed active photochemical processes, while the active photochemical processes at high PM$_{2.5}$ concentration were conductive to the production and accumulation of O$_3$ under a VOC-limited regime and a calm atmospheric condition including high temperature, moderately high relative humidity, and low wind speed, which in turn enhanced the conversions of SO$_2$ and NO$_2$ and the formation and accumulation of secondary sulfate and nitrate aerosols and further promoted the increase of PM$_{2.5}$ concentration and the deterioration of air pollution. Further analysis indicated that the daytime HONO concentration could be strongly negatively correlated with O$_3$ concentration in most of the double high pollution episodes, revealing the dominant role of HONO in O$_3$ formation during these pollution episodes. This study provides important field measurement-based evidence for understanding the significant contribution of daytime HONO to O$_3$ formation, and helps to clarify the formation and coexistence mechanisms of the double high-level O$_3$ and PM$_{2.5}$ pollution episodes.

**Keywords:** O$_3$; PM$_{2.5}$; HONO; double high pollution; secondary aerosol

## 1. Introduction

Shanghai is located in the eastern coast of the Yangtze River Delta (YRD), adjacent to the East Sea. As one of the largest metropolises in China, Shanghai has undergone rapid growth in economic and industrial development and urbanization over the past few decades. The increasing population density and world level industry concentrations

have led to escalating energy consumption and large emissions of pollutants [1], which substantially resulted in severe regional air pollution [2–5]. $O_3$ and $PM_{2.5}$ (particles with aerodynamic diameter $\leq 2.5$ μm) are the two most important pollutants in major Chinese city clusters such as YRD and have drawn increasing attention due to their impacts on visibility [6], human health [7], and climate change [8].

Previous studies have reported that the secondary aerosols, including secondary inorganic aerosols (SIA: $SO_4^{2-}$, $NO_3^-$, and $NH_4^+$) and secondary organic aerosols (SOA), are the main contributors to fine particulate in China. For example, SIA account for about 50% of $PM_{2.5}$ mass concentrations [9]. Due to complex photochemical reactions among primary pollutants such as $SO_2$, $NO_x$, and volatile organic compounds (VOCs) in the presence of solar radiation, a large number of SIA and SOA are generated, which constitute the main components of $PM_{2.5}$ and increase its concentration. As a secondary pollutant, $O_3$ in the troposphere is mainly produced by photochemical reactions, and $NO_X$ and VOCs generated from motor vehicles and industrial emissions are the main precursors of $O_3$ formation [10]. Generally, in the troposphere, $NO_2$ photolysis at $\lambda < 420$ nm generates NO and atomic oxygen O ($^3P$), then the formed O ($^3P$) interacts with $O_2$ to produce $O_3$. Once formed, $O_3$ readily reacts with the formed NO to regenerate $NO_2$, resulting in a null cycle when no other chemical oxidant species were involved. However, in the troposphere, there exist alternative oxidants (e.g., $HO_2$ and $RO_2$) that efficiently convert NO to $NO_2$, which results in the accumulation of $O_3$ [10], and therefore, $NO_2$ photolysis initiates the principal $O_3$-forming reaction in polluted air. Strong solar radiation, high temperature, and low wind speed are beneficial to the formation and accumulation of $O_3$ [10–13].

Some studies have been performed to investigate particulate matter pollution and/or $O_3$ pollution [13–16]. The concentration of $PM_{2.5}$ presents a distinct seasonal variation, generally showing high values in winter and low ones in summer [17–19]. High particulate matter pollution often corresponds to low concentrations of $O_3$ in cold seasons [20], and low airborne particulates to high $O_3$ pollution in warm seasons [21]. However, several studies have shown that some air pollution events present high levels of $O_3$ and particulates concurrently worldwide in recent years. For example, Awang et al. reported that ground-level $O_3$ concentrations were higher during high particulate events (HPE) than those during non-HPE in Malaysia [22]. Ding et al. found that biomass burning in agricultural activities caused high $PM_{2.5}$ and $O_3$ pollution in Nanjing, eastern YRD [17]. Tie et al. found the co-occurrence of high $PM_{2.5}$ and $O_3$ concentrations during late spring and early fall in eastern China [23]. Wang et al. investigated the primary characteristic of the high co-occurring concentrations of $O_3$ and $PM_{2.5}$ in YRD in the summer of 2013 [24]. However, the research on the double high-level $O_3$ and $PM_{2.5}$ pollution in the YRD and the corresponding occurrence mechanism is still very limited.

In addition, the surface $O_3$ concentration in China is increasing year by year [25], which is often considered to be closely related to the decrease in $PM_{2.5}$ concentration. This is because that the light extinction of aerosols weakens $O_3$ photochemistry, resulting in a significant reduction in the net chemical production of $O_3$, and therefore the increase of surface $O_3$ concentration in China was usually attributed to the high concentrations of $O_3$ aloft being entrained by turbulence from the top of the planetary boundary layer to the surface [26]. However, high particulate pollution episodes with high $O_3$ levels are observed from time to time in China. Furthermore, nitrous acid (HONO) has been recognized as a major precursor of tropospheric OH radical that initiates daytime atmospheric photochemistry [27,28]. For example, Kim et al. reported that HONO photolysis was the dominant OH source, contributing 80.4% of atmospheric primary OH production in the wintertime in Weld Country, Colorado [27]. A recent modelling study found that HONO photolysis acted as the dominant source for primary OH production with a contribution of more than 92% based on a winter field campaign conducted at a rural site of the North China Plain [28]. The primary OH production from the HONO photolysis enhances the formation of hydroperoxyl and organic peroxy radicals, and hence affects the formation of $O_3$. Some modeling studies showed that elevated levels of HONO considerably enhanced the ROx

(= OH + HO$_2$ + RO$_2$) concentrations and accelerated the RO$_X$ cycles across the Jing-Jin-Ji and coastal regions of China [29,30], and HONO photolysis during the daytime could significantly enhance O$_3$ formation in the polluted boundary layer [23,31,32]. However, up to now, few studies have been reported on the impact of HONO on O$_3$ formation during the specific double high-level PM$_{2.5}$ and O$_3$ pollution episodes.

In this study, based on a continuous field observation from 2014 to 2016 in Shanghai, a comprehensive analysis was carried out to investigate the characteristics and the formation and coexistence mechanisms of the double high-level PM$_{2.5}$ and O$_3$ pollution episodes. Meanwhile, the impact of daytime HONO on O$_3$ formation during the double high pollution episodes was also investigated. This study will help to further understand the characteristics and occurrence mechanism of the double high pollution episodes and the role of daytime HONO in O$_3$ formation in these pollution episodes.

## 2. Materials and Methods

### 2.1. Sampling Site Description

In this study, sampling was conducted from 3 April 2014, to 31 December 2016, in Pudong New Area Environmental Monitoring Station in Shanghai, China (31.23° N, 121.53° E) (Figure 1). Shanghai is one of the largest cities in the YRD region, and this site is representative of urban areas due to the high traffic density and residential and industrial emissions. Climatically, Shanghai has a subtropical monsoon climate, with four distinct seasons, that is, spring (March, April, and May), summer (June, July, and August), autumn (September, October, and November), and winter (December, January, and February). The annual average temperature (T), relative humidity (RH), and rainfall are about 16.3 °C, 78.4%, and 1182.8 mm, respectively.

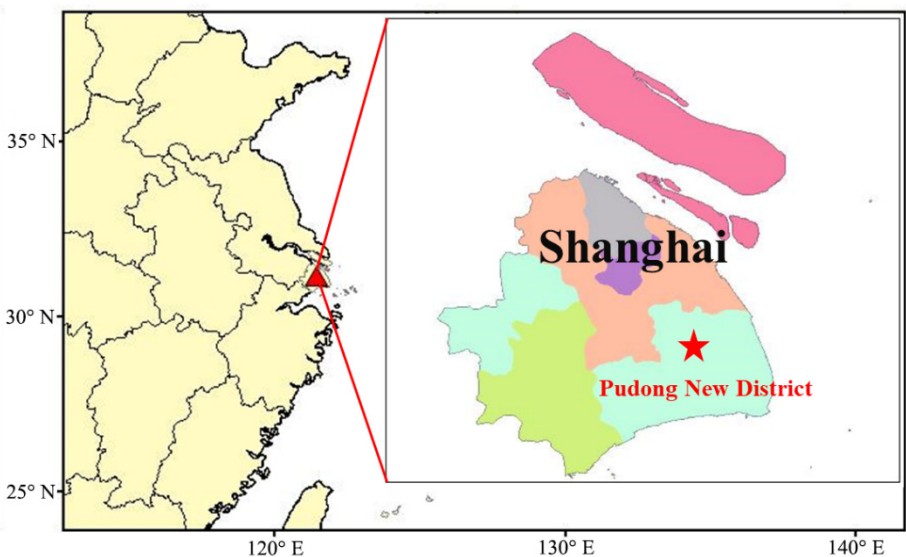

**Figure 1.** Map of the geographical location of the observation site.

### 2.2. Water-Soluble Inorganic Ion and Trace Gas Measurements

In this study, we acquired hourly mass concentrations of major water-soluble inorganic ions in PM$_{2.5}$ (NH$_4^+$, Na$^+$, K$^+$, Ca$^{2+}$, Mg$^{2+}$, SO$_4^{2-}$, NO$_3^-$, and Cl$^-$) and trace gases (HCl, HONO, SO$_2$, HNO$_3$, and NH$_3$) by using a continuously on-line Monitor for Aerosols and Gases in Ambient Air (MARGA, ADI 2080, Applikon Analytical B.B Corp., Netherlands) with a Teflon-coated PM$_{2.5}$ cyclone inlet. The detailed information about the MARGA instrument was given in previous studies [33,34]. Briefly, the instrument consists of a sampling box and an analytical box. The sampling box is comprised of a wet rotating denuder (WRD) for trace gas absorbing and a steam jet aerosol collector (SJAC) for particle collecting. The ambient air sampling for MARGA was performed at a flow rate of 1 m$^3$/h through the inlet by using a pump equipped with a mass flow controller. Gaseous species

were absorbed and dissolved into the liquid coated inter surface of the WRD, and then particles in the stream passed through the WRD and arrived in the SJAC to be condensed into droplets rapidly. After collecting for every hour, the two collected liquid samples were analyzed by ion chromatography with a C4 100 × 4 mm column for cations and a supp. 10–75 column for anions. MARGA was automatically calibrated by an internal calibration method using a lithium bromide solution. MARGA was also routinely calibrated by using a mixed cation and anion external standard solution in a routine time to ensure data quality.

### 2.3. $PM_{2.5}$ Mass Concentrations and Trace Gases

Hourly mass concentration of $PM_{2.5}$ and trace gas species, including $SO_2$, $O_3$, and $NO_X$, were continuously measured using a tapered element oscillating microbalance (TEOM 1405-D, Thermo Scientific Co., Waltham, MA, USA) and a series of gas analyzers (Ecotech EC9850B $SO_2$ automatic monitor for $SO_2$, 49i, and 42i gas analyzers for $O_3$ and $NO_X$, respectively, Thermo Scientific, USA). The high-resolution data during the sampling time were converted to hourly means. The analytical instruments were also routinely calibrated by using external standard gases, including daily zero check and weekly span calibrations to ensure data quality. It is worth noting that the TEOM measured mass has a strong temperature and RH dependence, and the effect is larger at larger concentrations, which leads to mass underestimation. Loss of semi-volatile compounds is the probable cause [35,36]. Therefore, the TEOM method underestimates $PM_{2.5}$ mass, and the underestimation is greater at high concentrations [35,36], which may mean that the actual $PM_{2.5}$ pollution in the double high pollution cases may be more serious than that revealed by the measured $PM_{2.5}$ mass data.

### 2.4. Meteorological Data

The hourly meteorological parameters, including T, RH, wind speed (WS), wind direction (WD), visibility (Vis), and precipitation, were obtained from the same environmental monitoring station. Meteorological datasets, including global solar radiation (SR) and boundary layer height (BLH), were derived from ADS (https://ads.atmosphere.copernicus.eu/cdsapp#!/home) (accessed on 9 August 2020) and ECMWF ERA-Interim reanalysis (https://apps.ecmwf.int/datasets/data/interim-full-daily/levtype=sfc/) (accessed on 10 August 2020) in a 0.125° × 0.125° grid, respectively.

## 3. Results and Discussion

### 3.1. Overview of $PM_{2.5}$ and Trace Gas Pollution in Shanghai

#### 3.1.1. Seasonal Behaviors

Figure 2 shows the monthly variations of $PM_{2.5}$, $O_3$, NO, $NO_X$, CO, and $SO_2$ in Shanghai during the period of April 2014–December 2016. As can be seen from Figure 2a, $O_3$ exhibits a distinguished seasonal variation during the observation period. In 2014, the curve of monthly $O_3$ concentration shows a sharp peak in late spring and a broad one in early autumn (a maximum in May and a secondary maximum in September). In 2015, $O_3$ concentration increased from January on and reached a relatively high level in May, which is a secondary maximum value of the whole year. Then, after a slight drop in June, it continued to increase until it reached a maximum value in September. The $O_3$ variation trend in 2016 is similar to that in 2014, while a slightly increasing trend appears in July rather than a drop in the same month of 2014. The observed seasonal characteristics from 2014 to 2016 in Shanghai are different from previous studies conducted in southern and northern China. For example, a summer minimum and an autumn maximum of $O_3$ were reported in Hong Kong [37], and an early summer (June) broad maximum was found in Beijing [38,39]. However, the seasonal cycle of $O_3$ concentrations exhibits similar trends as those found at Lin'an site on the southern edge of the YRD [40–42], in which $O_3$ concentration showed a maximum in May and a sharp drop in July. Shanghai is in the eastern region of YRD and verges on the East Sea, which is on the upwind of the southeast summer monsoon. Therefore, the local emission of Shanghai and the intense solar radiation

could be the main causes of $O_3$ formation in summer, resulting in a different seasonal cycle of $O_3$ compared to other inland cities of YRD. In fact, the $NO_X$ and CO (Figure 2c,f) show a relatively high level in summer (about 20 and 500 ppbv, respectively). As mentioned above, it is noticeable that a slight drop of $O_3$ concentration was found in summer compared to other seasons. Although high temperature and high solar radiation intensity in summer could promote the formation of $O_3$, the marine air transported from the Pacific Ocean from time to time due to the subtropical summer monsoon climate of Shanghai will not only increase precipitation (Table S1) but also bring strong winds, leading to a significant reduction of $O_3$ concentration from June to August. In addition, $O_3$ concentration shows a low level in winter. This could be attributed to weaker solar radiation, lower temperature, and high level of NO (Figure 2d), because the weaker solar radiation and lower temperature decreased atmospheric photochemical reactivities and the high NO level enhanced the chemical titration of $O_3$, and therefore $O_3$ production was suppressed.

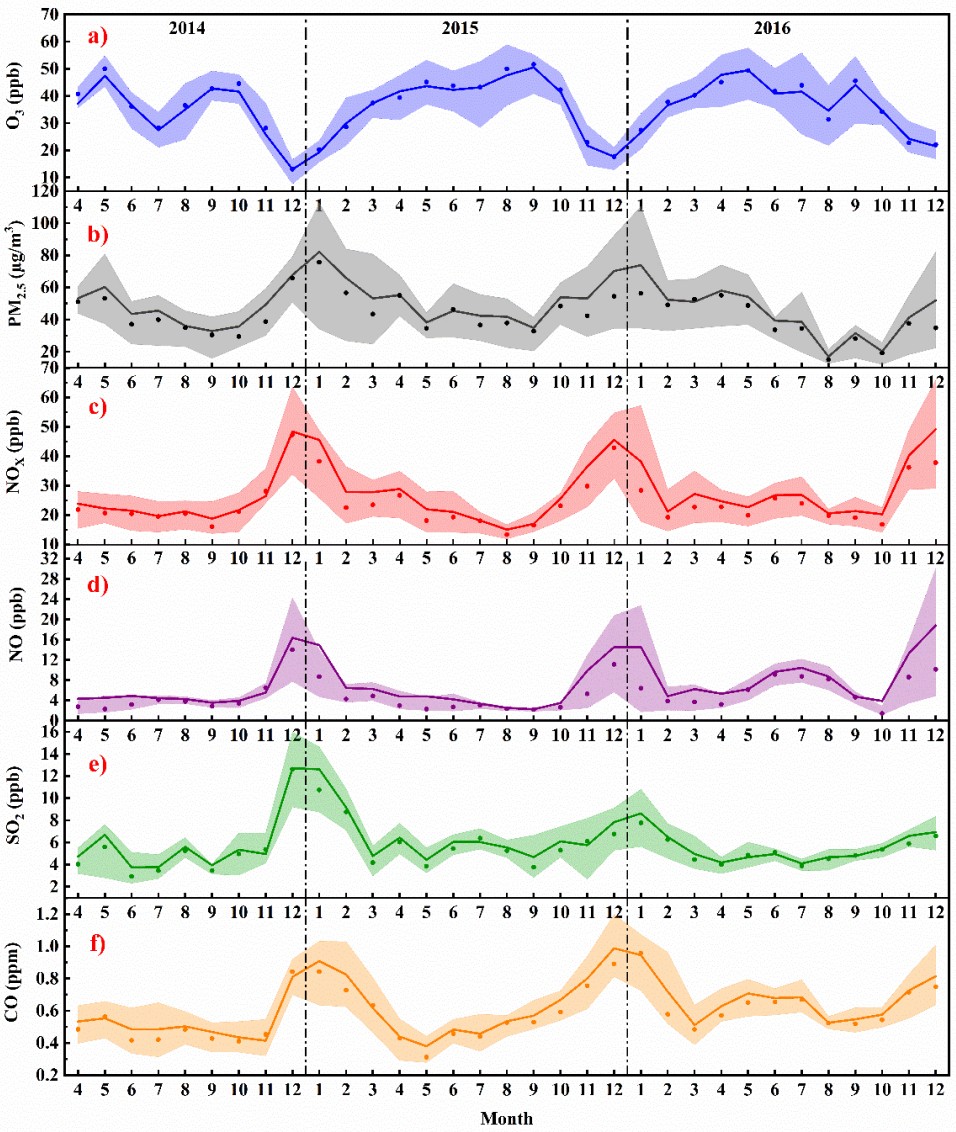

**Figure 2.** Seasonal variations of (**a**) $O_3$, (**b**) $PM_{2.5}$, (**c**) $NO_X$, (**d**) NO, (**e**) $SO_2$, and (**f**) CO. Bold solid lines are the median values, diamonds show the monthly averages, and thin solid lines represent percentiles of 75% and 25%.

Figure 2b gives the month-to-month concentration variations of fine particle $PM_{2.5}$ from 2014 to 2016. It shows an overall well-defined seasonal pattern with the maximum in winter (January) and the minimum in early autumn (September in 2014 and 2015) or late

summer (August in 2016). This distinct seasonal variation could be attributed to more fossil fuel combustion emissions in colder seasons and less in warmer seasons, and the changes of ambient vertical mixing processes and deposition in different seasons. On the one hand, local fossil fuel combustion emissions increased in winter. On the other hand, as we know, the increased fossil fuel combustion emissions during the winter heating period in the North China Plain enhanced the particulate matter and other trace gases concentrations of YRD region via long-range transport [15,43]. Atmospheric conditions such as lower mixing layer height and lower temperature would decrease the horizontal and vertical mixing processes. Furthermore, deposition would have a distinguished seasonal variation, because heavy precipitation in summer led to wet-deposition and high soil humidity and the growth of deciduous plants promoted the dry deposition of particulate matter in warmer seasons [44]. The relatively low concentration of $PM_{2.5}$ in summer might be partly due to the increase of mixing boundary layer height and the more active vertical process [19]. It is worth noting that $PM_{2.5}$ concentration showed a strong change in the monthly curve from 2014 to 2016. For instance, an apparent drop in February could be found both in 2015 and 2016, as well as a sudden drop in October 2016. The concentration drops of $PM_{2.5}$ together with other primary pollutants (i.e., $SO_2$, CO, $NO_X$) in February were mainly due to the winter break of the Chinese Spring Festival and the ban on setting off fireworks in Shanghai during the festival. In addition, it can be clearly seen that the month-by-month change patterns of $PM_{2.5}$ from 2014 to 2016 are similar to that of $NO_X$, indicating that vehicle emissions and fuel combustion emissions have important impacts on $PM_{2.5}$ concentration.

### 3.1.2. Distinguishment of Double High-Level $O_3$ and $PM_{2.5}$ Episodes

Figure 3a–c displayed the scatter plots of $O_3$-$NO_X$, CO-$NO_X$, and $PM_{2.5}$-$O_3$ relationships during the whole observation period, respectively. Moreover, to analyze the influence of related factors, the dataset points were color-coded with different parameters such as ambient temperature and $O_3$ concentration. Figure 3a illustrated the scatter plots of $O_3$ and $NO_X$ measured at the observation site with temperature during the whole observation period. As reported in previous studies [17,19], $O_3$ showed an overall negative correlation with $NO_X$ based on the whole dataset. The colored scatters showed that the negative correlation mainly occurred at lower ambient temperature, indicating a titration effect of freshly emitted NO with $O_3$ in cold seasons and/or an impact of high RH, especially at night. In contrast, a positive correlation between $O_3$ and $NO_X$ was found at higher temperatures (>25 °C) and a moderate level of $NO_X$ concentration (<100 ppbv), which mainly occurred in the daytime of warmer seasons. This result possibly suggested an intensely active photochemical reaction of $O_3$ production in summer, which led to a strong seasonal variation pattern, as shown in Figure 2a. In addition, it is worth noting that there is usually a negative correlation between T and RH, and $O_3$ can react with water vapor to some extent or dissolve in particles containing liquid water, which would also cause low $O_3$ concentration at low T and high RH. In other words, low T and high RH are not conducive to the formation and accumulation of $O_3$.

Generally, CO has a good correlation with VOCs and plays a similar role as VOCs in photochemical $O_3$ production [12], and thus in this study CO was chosen as the reference species for VOCs, because VOCs were not measured. A scatter plot of CO and $NO_X$ color-coded with $O_3$ concentration was provided by Figure 3b. It is found that high concentration of $O_3$ was usually associated with high CO/$NO_X$ ratio, and an increase of CO always led to higher $O_3$ concentration when $NO_X$ concentration was lower than 75 ppbv, while $NO_X$ reversed. This relationship based on CO-$O_3$-$NO_X$ suggested a VOC-limited regime of $O_3$ formation in urban Shanghai. The inset in Figure 3b also specifically illustrated the VOC-limited region of our observation site. This result is similar to the previous studies by Ding et al. and Chen et al., who also found a VOC-limited regime in urban sites of Nanjing and Hangzhou in YRD, respectively [2,17].

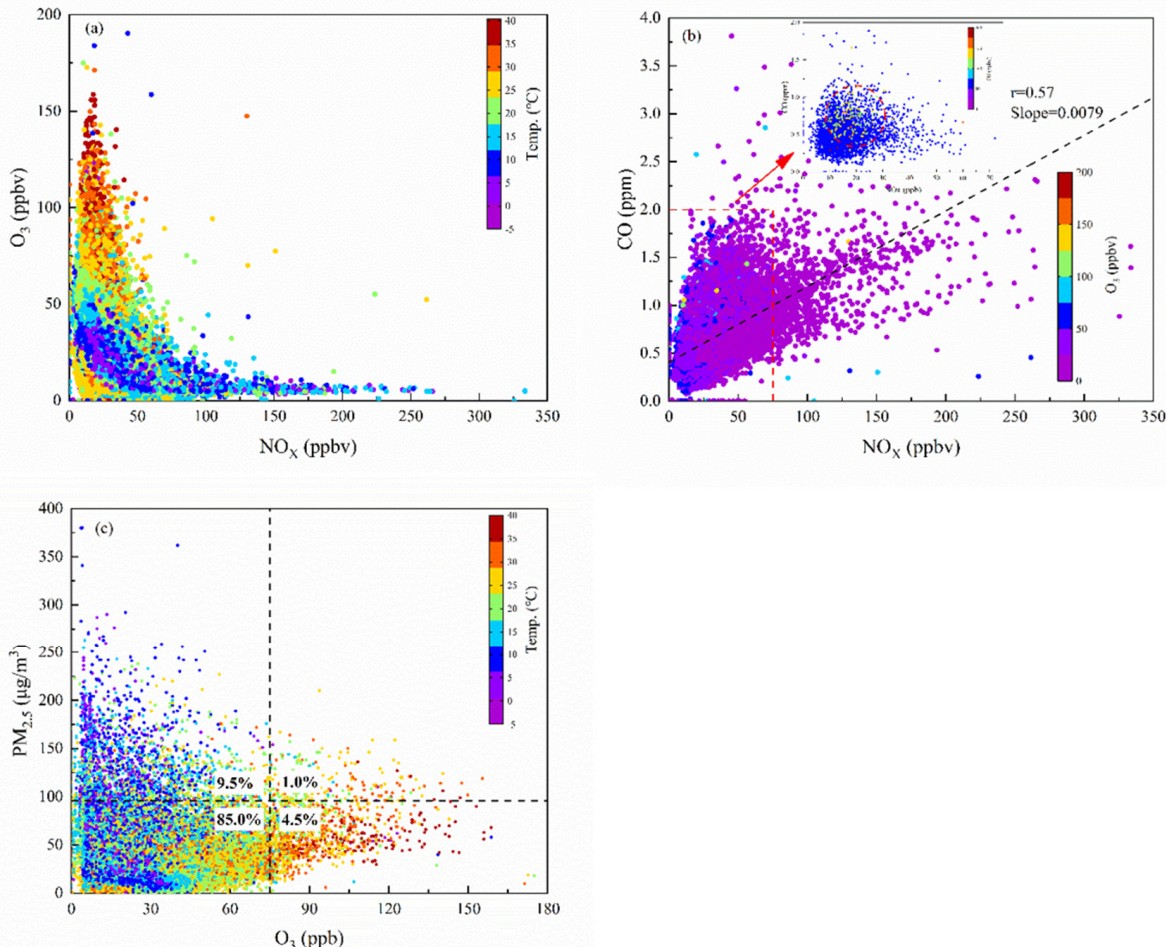

**Figure 3.** Scatter plots of selected parameters during the whole observation period. (**a**) $O_3$-$NO_X$ color-coded with air temperature, (**b**) CO-$NO_X$ color-coded with $O_3$, and (**c**) $PM_{2.5}$-$O_3$ color-coded with air temperature.

The color-coded scatter plots in Figure 3c showed overall high $O_3$ concentrations at high temperature and high $PM_{2.5}$ levels at the reverse situation. The relationship of $PM_{2.5}$ and $O_3$ indicated a significantly positive correlation under higher temperature and a negative correlation under low ambient temperature. Similar results were found in Nanjing and Hangzhou by previous reports [2,17,19]. The positive correlation suggested that a high concentration of $O_3$ was favorable to the formation of secondary aerosols in $PM_{2.5}$ at high temperatures. The anti-correlation for cold air might be mainly attributed to the strong titration effect of NO in cold seasons and the dimming effect of high $PM_{2.5}$ mass concentration. On the one hand, the increasing slope under high ambient temperatures probably reflected the enhanced formation of secondary particulates, which were closely related to the high conversion rate of $SO_2$ to sulfate in the presence of high-level $O_3$ and high solar radiation [45]. Therefore, it led to the mixed pollution of high levels of fine particulate and $O_3$ in warmer periods. In China, particulate pollution events were often defined as $PM_{2.5}$ daily average concentration exceeded the class-II of the Chinese National Ambient Air Quality Standard (GB3095–2012, 75 $\mu g/m^3$) [46]. Some studies also defined a specific value as high particulate event according to the actual situation. For instance, Huang et al. suggested that days with $PM_{10}$ levels exceeding 56 $\mu g/m^3$ were defined as high PM days [47]. On the other hand, previous studies often defined $O_3$ pollution based on the Chinese National Ambient Air Quality Standard (GB3095–2012), which suggested the limited value of the second grade daily maximum hourly concentration (200 $\mu g/m^3$, 93 ppb) and daily hourly concentration (160 $\mu g/m^3$, 75 ppb). In this study, the scatter points herein were divided into four regions with the species concentrations of 96 $\mu g/m^3$ and 75 ppb for $PM_{2.5}$ and $O_3$, respectively (marked with black-dotted line). The data with

moderate concentration levels for both $PM_{2.5}$ and $O_3$ at relatively lower temperatures during the entire three years accounted for about 85.0% of total sampling data points, which indicated that the double low $O_3$ and $PM_{2.5}$ concentrations mainly occurred in the seasons with moderate temperature. The data with high-$PM_{2.5}$–low-$O_3$ concentration at low temperature and the high-$O_3$–low-$PM_{2.5}$ concentration at high temperature accounted for the second and third proportions (9.5% and 4.5%, respectively) during the whole period. The last but most noticeable was the cases with data of double high $O_3$ and $PM_{2.5}$ concentrations, accounting for about 1.0% of total sampling data points. It should be pointed out that this pollution mainly occurred between April and September in Shanghai, but it did not easily occur in early spring, late autumn, or winter in Shanghai. The need for active photochemical processes may be an important reason for the occurrence of this pollution, even in the presence of high concentrations of $PM_{2.5}$.

3.1.3. Characteristics of Double Low-Level $O_3$ and $PM_{2.5}$ Periods

In the four regions in Figure 3c, double low-level $O_3$ and $PM_{2.5}$ episodes frequently occurred during the entire observation period (Figure 2a,b). Herein, three typical cases with double low-level $O_3$ and $PM_{2.5}$ were selected for analysis and comparison, and the corresponding meteorological factors and chemical species concentrations were summarized in Table S2. The three cases occurred from November 26 to November 27, 2014 (denoted as Case 1), from 8 February to 10 February 2015 (denoted as Case 2), and from 12 February to 22 February 2016 (denoted as Case 3), during the three-year observation period, respectively.

In the three cases, the average concentration of $PM_{2.5}$ was $43.5 \pm 25.4$ $\mu g/m^3$, exceeding the first grade of the Chinese National Ambient Air Quality Standard for daily averaged $PM_{2.5}$ concentration (GB3095–2012, 35 $\mu g/m^3$). The average mass concentrations of $PM_{2.5}$ in Case 2 and Case 3 were $46.9 \pm 30.9$ and $45.2 \pm 24.3$ $\mu g/m^3$, respectively, and the values were about 1.6 times higher than that of Case 1 ($28.9 \pm 15.6$ $\mu g/m^3$). The $PM_{2.5}$ average mass concentrations in these cases were lower than that in the previous study, which was $66 \pm 29$ $\mu g/m^3$ in non-haze days [5]. The average mass concentration of $O_3$ was $66.5 \pm 26.8$ $\mu g/m^3$, and it was lower than the second grade of the Chinese ambient air quality standard (GB3095–2012, 160 $\mu g/m^3$ per hour for $O_3$ for second grade). The average mass concentrations of SIA and total water-soluble inorganic ions (TWSI) were $26.43 \pm 16.70$ and $29.13 \pm 17.65$ $\mu g/m^3$, which accounted for 6% and 65% of $PM_{2.5}$, respectively, and SIA accounted for 89% of the TWSI during double low-level $O_3$ and $PM_{2.5}$ periods. Both TWSI and SIA during double low-level periods had a good positive correlation with $PM_{2.5}$, respectively, and their correlation coefficient (r) values with $PM_{2.5}$ were 0.95 and 0.96, respectively, indicating that the TWSI and SIA contributed significantly to $PM_{2.5}$. The average mass concentrations of $NO_3^-$, $SO_4^{2-}$, and $NH_4^+$ were $11.97 \pm 9.58$, $7.63 \pm 4.00$, and $6.83 \pm 4.44$ $\mu g/m^3$ during the double low-level periods, respectively. In the three cases, the average mass concentration of $NO_3^-$ obviously exceeded that of $SO_4^{2-}$, indicating high nitrate pollution in Shanghai, which is consistent with the previous studies [48,49]. The occurrence of high nitrate pollution should be attributed to the reduction of $SO_2$, the increase of $NO_X$ emission due to $SO_2$ emission control strategies of China, and the rapid increase of $NO_X$ emitted from dramatically increased vehicle population and vigorously developed industries [48,49].

*3.2. General Characteristic of Double High-Level $O_3$ and $PM_{2.5}$ Episodes*

In this study, we defined the double high-level $O_3$ and $PM_{2.5}$ pollution episodes as the condition that $PM_{2.5}$ and $O_3$ concentrations consecutively exceeded 96 and 160 $\mu g/m^3$ (75 ppb) for an 8-h period or longer, respectively, and hence a total of nine typical cases were found during the whole field observation period. These mixed pollution cases mainly occurred between April and September in Shanghai, but it did not easily occur in early spring, late autumn, or winter in Shanghai, showing distinct seasonal characteristics. To further understand the pollution characteristics, main cause, and occurrence mechanism of

the mixed pollution episodes with double high-level $O_3$ and $PM_{2.5}$, we selected six typical pollution episodes for detailed case studies.

3.2.1. Mixed Pollution Episodes with Double High-Level $PM_{2.5}$ and $O_3$

Figure 4 and Table 1 described the temporal variations and summary statistics of key pollutants and meteorological parameters during the six case periods. As shown in Figure 4, $PM_{2.5}$ usually started to increase at 6:00 a.m. and reached the highest level (even up to 333 $\mu g/m^3$ in case 3) at 12:00–15:00 in the afternoon. Then, it gradually decreased and dropped to the lowest point at midnight. The average concentration of $PM_{2.5}$ in mixed pollution episodes was 92.1 $\pm$ 32.1 $\mu g/m^3$, exceeding the second grade of the Chinese National Ambient Air Quality Standard for daily average $PM_{2.5}$ mass concentration (GB3095-2012, 75 $\mu g/m^3$). The value was 2.12 times higher than that of the double low-level days. As for $O_3$, the average concentration remained at a high level, even reaching ~169.0 $\mu g/m^3$ in case 6 (11–12 May 2016). As shown in Figure 4a, $O_3$ concentration was lower in the morning (6:00–8:00 a.m.) and evening (16:00–20:00 p.m.) but peaked in the afternoon (13:00–15:00 p.m.), which corresponded well to the traffic rushes and active photochemical period of one day. The morning (6:00–8:00 a.m.) and evening (16:00–20:00 p.m.) traffic rushes increased the concentrations of $NO_2$ and NO, and thus the lower $O_3$ concentrations at the two stages were mainly due to relatively weak solar radiation and NO titration. The $O_3$ concentration rapidly increased from 8:00 a.m. and reached the maximum value at 13:00–15:00 p.m., which was mainly due to the photolysis of accumulated $NO_2$ and the increasing active photochemical processes. Then, the $O_3$ was affected by vertical mixing, horizontal diffusion, and various physical and chemical consumptions in the atmosphere, and its concentration decreased until nighttime. The above results indicated that active photochemical reactions still occurred in the daytime under high $PM_{2.5}$ concentration, in which the formation and accumulation of various oxidizing species including $O_3$ enhanced the conversions of $NO_2$ and $SO_2$ into $NO_3^-$ and $SO_4^{2-}$ and further promoted the evolution of fine particulate matter pollution.

In addition, $O_3$ concentration has a good correspondence with the temperature peak and RH trough. This is because that high temperature often means strong solar radiation. Moreover, $O_3$ diurnal variation between daytime and nighttime could be generally distinguished by solar radiation. Similar diurnal variation trends were also found in Malaysia [22]. It is noticeable that a rapid increase in $PM_{2.5}$ mass concentration during these cases was often accompanied by the rapid rise of $O_3$ and kept for several hours in the daytime, and sometimes the two pollutants even peaked simultaneously. On the one hand, this result indicated that active photochemical processes still occurred and $O_3$ could be generated and accumulated under high-level $PM_{2.5}$. On the other hand, the occurrence of heavy fine particulate pollution can be closely associated with the active photochemical processes. The average mass concentration of $O_3$ was 151.7 $\pm$ 76.8 $\mu g/m^3$ on the double high pollution days, which was 2.28 times higher than that of the double low days (66.5 $\pm$ 26.5 $\mu g/m^3$). This result suggested strong atmospheric oxidizing capacity during the double high pollution cases, which would promote the formation of SIA and thus further enhance the concentration of $PM_{2.5}$. This study further confirmed that high fine particulate pollution and high ozone pollution can occur simultaneously under suitable atmospheric conditions.

**Table 1.** Summary of meteorological factors and chemical species data in the double high-level $O_3$ and $PM_{2.5}$ pollution cases.

| Parameter/Specaies | Case 1 | Case 2 | Case 3 | Case 4 | Case 5 | Case 6 | Average |
|---|---|---|---|---|---|---|---|
| $PM_{2.5}$ ($\mu g/m^3$) | 93.7 ± 18.4 | 99.5 ± 39.3 | 95.2 ± 10.6 | 86.4 ± 34.6 | 97.4 ± 38.8 | 78.8 ± 25.9 | 92.1 ± 32.1 |
| $O_3$ ($\mu g/m^3$) | 132.4 ± 63.6 | 122.9 ± 90.0 | 147.7 ± 104.3 | 143.8 ± 89.5 | 169.0 ± 57.1 | 155.8 ± 73.9 | 151.7 ± 76.8 |
| $O_3$-8 h ($\mu g/m^3$) | 145.4 ± 52.0 | 140.8 ± 74.5 | 169.7 ± 85.1 | 164.3 ± 73.6 | 174.1 ± 43.7 | 169.7 ± 54.6 | 146.8 ± 67.1 |
| $SO_2$ ($\mu g/m^3$) | 23.25 ± 7.67 | 17.33 ± 10.69 | 26.92 ± 33.95 | 30.33 ± 4.71 | 20.36 ± 7.18 | 19.52 ± 6.02 | 22.15 ± 13.70 |
| $NO_2$ ($\mu g/m^3$) | 36.50 ± 9.58 | 38.50 ± 11.99 | 53.54 ± 24.72 | 40.67 ± 10.03 | 35.78 ± 16.67 | 37.11 ± 19.00 | 39.04 ± 17.40 |
| CO ($\mu g/m^3$) | 789.17 ± 104.77 | 851.08 ± 233.43 | 948.00 ± 294.60 | 791.54 ± 106.86 | 1154.06 ± 386.35 | 993.81 ± 220.64 | 983.95 ± 312.48 |
| NOx ($\mu g/m^3$) | 40.54 ± 11.92 | 45.17 ± 18.69 | 68.33 ± 52.56 | 46.54 ± 11.06 | 38.89 ± 18.53 | 48.21 ± 33.56 | 46.07 ± 28.32 |
| $K^+$ ($\mu g/m^3$) | 1.09 ± 0.13 | 1.58 ± 0.47 | 1.49 ± 0.25 | 1.06 ± 0.13 | 0.58 ± 0.29 | 0.47 ± 0.17 | 0.88 ± 0.49 |
| $Ca^{2+}$ ($\mu g/m^3$) | 0.24 ± 0.10 | 0.23 ± 0.14 | 0.22 ± 0.09 | 0.46 ± 0.07 | 0.25 ± 0.10 | 0.25 ± 0.09 | 0.27 ± 0.12 |
| $Na^+$ ($\mu g/m^3$) | 0.31 ± 0.04 | 0.35 ± 0.07 | 0.28 ± 0.04 | 0.32 ± 0.03 | 0.00 ± 0.00 | 0.22 ± 0.05 | 0.19 ± 0.14 |
| $Mg^{2+}$ ($\mu g/m^3$) | 0.09 ± 0.05 | 0.14 ± 0.05 | 0.11 ± 0.04 | 0.03 ± 0.04 | 0.06 ± 0.04 | 0.06 ± 0.03 | 0.08 ± 0.05 |
| $Cl^-$ ($\mu g/m^3$) | 1.02 ± 0.52 | 0.27 ± 0.49 | 0.00 ± 0.00 | 1.75 ± 0.40 | 0.98 ± 0.68 | 0.68 ± 0.44 | 0.81 ± 0.70 |
| $NO_3^-$ ($\mu g/m^3$) | 20.85 ± 7.68 | 20.72 ± 6.97 | 12.69 ± 5.74 | 13.45 ± 8.89 | 25.59 ± 14.31 | 17.53 ± 9.04 | 19.89 ± 11.43 |
| $SO_4^{2-}$ ($\mu g/m^3$) | 20.53 ± 5.53 | 27.18 ± 13.34 | 26.73 ± 5.05 | 22.30 ± 10.45 | 14.53 ± 4.59 | 14.98 ± 5.23 | 18.97 ± 8.71 |
| $NH_4^+$ ($\mu g/m^3$) | 12.97 ± 2.58 | 15.92 ± 5.84 | 12.98 ± 1.38 | 13.46 ± 5.86 | 13.91 ± 6.39 | 11.23 ± 4.44 | 13.27 ± 5.27 |
| HONO (ppbv) | 1.49 ± 0.55 | 2.18 ± 1.20 | 2.50 ± 1.49 | 4.74 ± 1.50 | 1.56 ± 0.79 | 2.14 ± 2.00 | 2.22 ± 1.64 |
| HONO (ppbv) daytime | 1.15 ± 0.38 | 2.00 ± 1.48 | 2.21 ± 1.78 | 4.21 ± 1.81 | 1.16 ± 0.75 | 1.48 ± 1.60 | 1.78 ± 1.60 |
| SIA ($\mu g/m^3$) | 54.35 ± 10.43 | 63.83 ± 24.19 | 52.40 ± 5.44 | 49.21 ± 21.65 | 54.03 ± 24.72 | 43.73 ± 17.09 | 52.13 ± 20.62 |
| TWSI ($\mu g/m^3$) | 57.09 ± 10.86 | 66.39 ± 25.08 | 54.50 ± 5.45 | 52.84 ± 21.78 | 55.91 ± 25.52 | 45.41 ± 17.58 | 54.36 ± 21.22 |
| SIA/TWSI | 0.95 ± 0.01 | 0.96 ± 0.01 | 0.96 ± 0.01 | 0.92 ± 0.03 | 0.97 ± 0.01 | 0.96 ± 0.01 | 0.96 ± 0.02 |
| TWSI/$PM_{2.5}$ | 0.62 ± 0.10 | 0.68 ± 0.04 | 0.58 ± 0.07 | 0.61 ± 0.04 | 0.55 ± 0.11 | 0.56 ± 0.07 | 0.58 ± 0.09 |
| SIA/$PM_{2.5}$ | 0.59 ± 0.09 | 0.65 ± 0.04 | 0.56 ± 0.07 | 0.56 ± 0.05 | 0.51 ± 0.13 | 0.54 ± 0.07 | 0.56 ± 0.08 |
| $NO_3^-/SO_4^{2-}$ | 1.13 ± 0.59 | 0.98 ± 0.53 | 0.52 ± 0.28 | 0.65 ± 0.42 | 1.64 ± 0.67 | 1.20 ± 0.68 | 1.17 ± 0.71 |
| SOR | 0.52 ± 0.09 | 0.68 ± 0.15 | 0.58 ± 0.09 | 0.82 ± 0.06 | 0.50 ± 0.11 | 0.65 ± 0.14 | 0.60 ± 0.15 |
| NOR | 0.30 ± 0.08 | 0.29 ± 0.09 | 0.16 ± 0.06 | 0.19 ± 0.10 | 0.34 ± 0.15 | 0.27 ± 0.13 | 0.28 ± 0.13 |
| Temp (ºC) | 19.2 ± 4.8 | 21.4 ± 3.9 | 26.3 ± 3.8 | 29.6 ± 3.1 | 19.9 ± 3.8 | 20.9 ± 3.6 | 22.0 ± 5.1 |
| RH (%) | 72.2 ± 15.2 | 82.1 ± 13.4 | 65.7 ± 13.2 | 65.1 ± 11.6 | 53.2 ± 18.7 | 61.8 ± 12.9 | 63.6 ± 17.4 |
| WS (m/s) | 2.9 ± 1.2 | 2.7 ± 1.6 | 3.3 ± 1.4 | 1.0 ± 0.3 | 1.5 ± 0.6 | 1.6 ± 0.7 | 1.9 ± 1.2 |
| Vis (km) | 7.5 ± 2.7 | 5.1 ± 1.2 | 7.0 ± 1.5 | 10.1 ± 4.0 | 9.5 ± 7.2 | 9.0 ± 5.6 | 8.3 ± 5.3 |

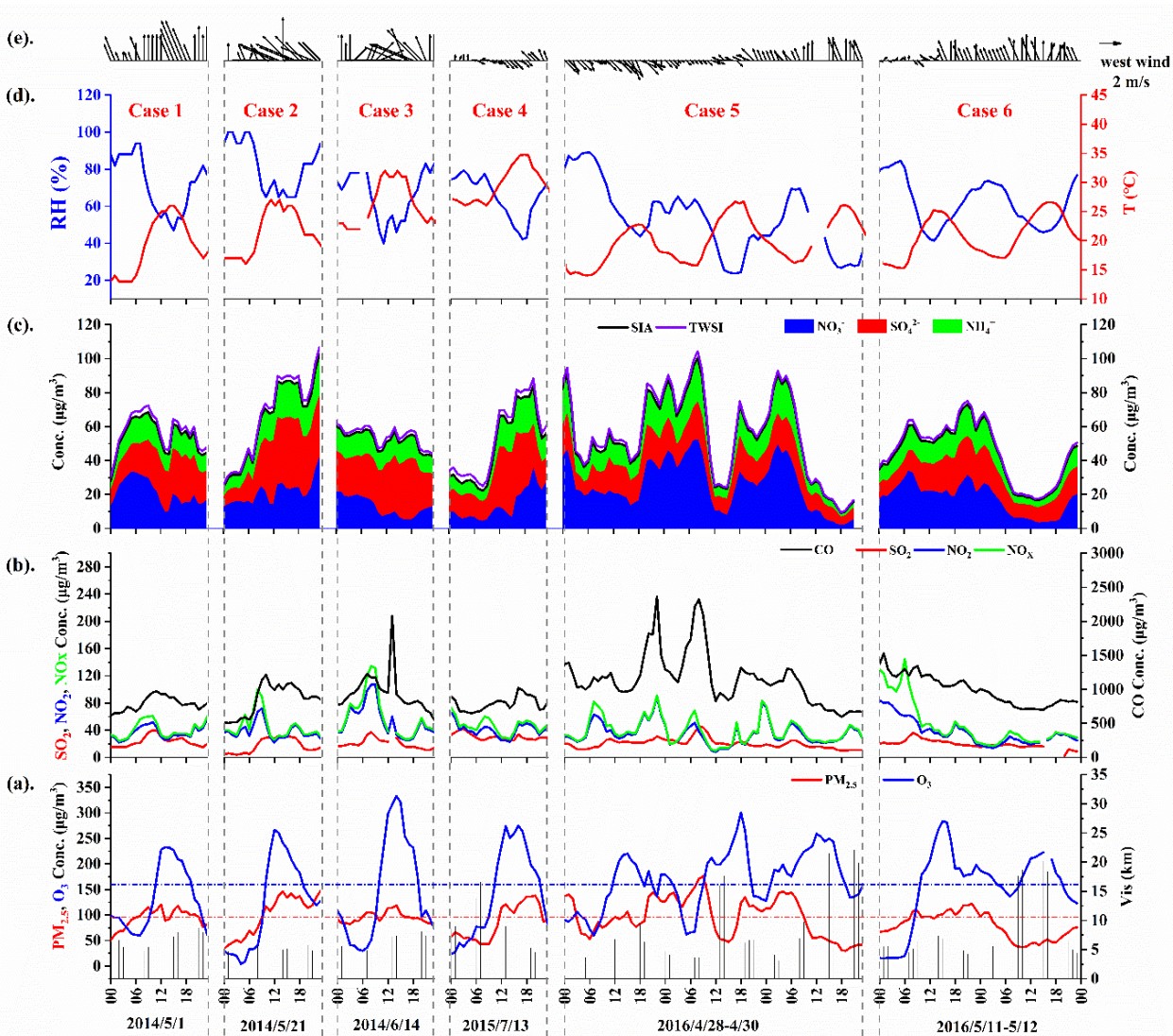

**Figure 4.** Time series of meteorological parameters and chemical species in double high-level $O_3$ and $PM_{2.5}$ pollution cases. (**a**) $PM_{2.5}$ ($\mu g/m^3$), $O_3$ ($\mu g/m^3$) and visibility (Vis, km); (**b**) $SO_2$, $NO_2$, $NO_X$ and CO ($\mu g/m^3$); (**c**) $NO_3^-$, $SO_4^{2-}$, $NH_4^+$, SIA and TWSI ($\mu g/m^3$); (**d**) temperature (T, °C) and relative humidity (RH, %); (**e**) wind direction and wind speed (m/s).

### 3.2.2. Characteristic of Water-Soluble Secondary Inorganic Ions in $PM_{2.5}$ during the Double High-Level $PM_{2.5}$ and $O_3$ Pollution Episodes

Chemical composition of fine particle matter is complex, including sulfate, nitrate, ammonium, organic carbon, inorganic carbon, and metal elements, among which secondary inorganic aerosol ions are important components [48,50]. In view of the high atmospheric oxidizing capacity during the double high pollution episodes, herein we mainly take SIA into consideration. In this study, the average mass concentration of SIA in $PM_{2.5}$ was $52.13 \pm 20.62$ $\mu g/m^3$ during the double high pollution episodes, which was about twice as much as that of the double low periods (Table 1 and Table S2). SIA accounted for 56% of the $PM_{2.5}$ during the double high pollution episodes, which was higher than that in the previously reported haze events in Shanghai [48,50,51]. The value was also higher than those in some European countries where the ratios were 42–48% and 22.4% $\pm$ 16% in Netherland and southern Italy, respectively [52,53]. This result indicated the high pollution characteristics of SIA and revealed that the secondary inorganic species were still one of the most predominant components of $PM_{2.5}$ during the double high pollution periods.

Figure 4c indicated the mass concentrations of TWSI and SIA were intensely close in all cases, and their trends matched perfectly with $PM_{2.5}$. Both the TWSI and SIA had a good positive correlation with $PM_{2.5}$, for which the overall correlation coefficients (r) were 0.942 and 0.943, respectively. From Figure 4c, during the double high pollution cases, the hourly mass concentrations of $PM_{2.5}$ increased with a rapid rise of TWSI and SIA, which even showed the same peaks and troughs simultaneously. This result indicated that TWSI and SIA played a crucial role in the formation of $PM_{2.5}$, which is consistent with the previous studies [48].

The average mass concentrations of $NH_4^+$, $SO_4^{2-}$, and $NO_3^-$ were $13.27 \pm 5.27$, $18.97 \pm 8.71$, and $19.89 \pm 11.43$ $\mu g/m^3$ during the double high pollution cases (Table 1), respectively, which were 1.94, 2.60, and 1.66 times higher than those of the double low episodes ($NH_4^+$: $6.83 \pm 4.44$ $\mu g/m^3$, $SO_4^{2-}$: $7.63 \pm 4.00$ $\mu g/m^3$, and $NO_3^-$: $11.97 \pm 9.58$ $\mu g/m^3$) (Table S2), respectively. This suggested the enhanced conversions of $SO_2$ and $NO_2$ and the enhanced formation of secondary $SO_4^{2-}$ and $NO_3^-$ [51], and then made a significant contribution to the high concentration of $PM_{2.5}$. The sulfur oxidation ratio (SOR) and the nitrate oxidation ratio (NOR) have usually been used as indicators of secondary transformation of gaseous species in the atmosphere [54], i.e., SOR = $n(SO_4^{2-})/(n(SO_4^{2-})$ + $n(SO_2))$, NOR = $n(NO_3^-)/(n(NO_3^-) + n(NO_2))$ (n refers to the molar concentration). Meanwhile, as reported in previous studies, SOR values were less than 0.1 in primary emissions and higher than 0.1 when sulfate was produced through the secondary oxidation processes of $SO_2$ [55]. High SOR and NOR indicate that the secondary oxidation processes are obvious. The average values of SOR and NOR in the six double high pollution cases were $0.60 \pm 0.15$ and $0.28 \pm 0.13$, respectively (Table 1), and the average values of SOR were generally higher than those of NOR. These values were higher than those observed in Shanghai in previous studies. For instance, Zhou et al. determined that the average values of SOR and NOR were 0.257 and 0.101 in 2011, respectively [56]. Hu et al. found that the average values of SOR and NOR on haze days in December 2013 were $0.32 \pm 0.09$ and $0.19 \pm 0.13$, respectively [57]. Meanwhile, the SOR and NOR values in the double high pollution episodes were also higher than those in the double low cases, which were $0.48 \pm 0.24$ and $0.19 \pm 0.10$ for SOR and NOR, respectively (Table S2). Therefore, the high oxidation rates of $SO_2$ and $NO_2$ during the double high pollution episodes would lead to the formation of high levels of $SO_4^{2-}$ and $NO_3^-$ and thus led to further evolution of fine particulate pollution. The SOR value in Case 4 (Table 1, Figure S2) was $0.82 \pm 0.06$, the highest value among all the cases. The T and RH in Case 4 were $29.6 \pm 3.1$ °C and $65.1 \pm 11.6\%$, respectively, which were advantageous conditions for $SO_2$ to be transformed into $SO_4^{2-}$ via a heterogeneous process [58]. Meanwhile, the relatively lower RH did not favor the interaction between $O_3$ and water molecules, but favored the accumulation of $O_3$ in Case 4, and therefore atmospheric oxidizing capacity was enhanced, which also accelerated the conversion of $SO_2$ to $SO_4^{2-}$. For NOR, in Case 1, 2, 5, and 6, NOR values were clearly higher than those in other cases. Relatively low T and/or relatively high RH were common features of Case 1, 2, 5, and 6, while low T and high RH favored a shift from the gas phase as nitric acid to the particulate phase as ammonium nitrate, which favored the formation of $NO_3^-$. Moreover, the temperatures in Case 3 and Case 4 were $26.3 \pm 3.8$ and $29.6 \pm 3.1$ °C, respectively, which were the higher temperatures in the six pollution cases, and hence the NOR values in these two cases were the lowest two values among all cases.

Figure 4c also showed the trends of individual secondary inorganic ions in the six pollution cases, from which we can find the different variations of SIA formation. On the whole, high temperature and high levels of $SO_2$, $O_3$, and $NO_2$ were favorable for the formation of $SO_4^{2-}$, whereas the impact of RH on the $SO_4^{2-}$ formation was complex (Figure 4). The correlations of $SO_4^{2-}$ concentration with RH were extremely highly negative in all the mixed pollution cases except Cases 5 and 6 (r = 0.290, −0.040, respectively), with the highest value r = 0.923 (p < 0.01) in Case 4. The different correlation coefficients may suggest different formation pathways of atmospheric sulfate aerosols under different RH,

which led to different correlations between RH and $SO_4^{2-}$. High temperature not only promoted the reactions of $SO_4^{2-}$ formation, but also often reflected strong solar radiation, and thus led to active atmospheric photochemistry, which further promoted the conversion of $SO_2$ and the formation of $SO_4^{2-}$ [59]. The conspicuously positive correlations of $SO_4^{2-}$ concentrations with temperature were found in all cases, among which the highest value was r = 0.923 (p < 0.01) in Case 4. Moreover, the high concentration of $O_3$ meant strong atmospheric oxidizing capacity, while the existence of high concentration of $NO_2$ might also promote the conversion of $SO_2$ [60], and thus all of these promoted the conversion of $SO_2$ and the formation of $SO_4^{2-}$. The effect of temperature and RH on nitrate formation was also complex. The correlation of $NO_3^-$ with temperature was showing negative in Case 1, 3, 5, and 6 (r = −0.512, −0.712, −0.625, and −0.550, respectively), while nonsignificant correlation showed in Case 2 (r = 0.287) and positive correlation in Case 4 (r = 0.577). For RH, positive correlations of $NO_3^-$ with RH were found in Case 1, 3, 5, and 6 (r = 0.560, 0.565, 0.447, and 0.467, respectively) and negative correlations in Case 4 (r = −0.406) and Case 2 (r = −0.303). The different correlation coefficients between $NO_3^-$ and T or RH suggested the complex formation pathways of atmospheric nitrate aerosol under different T and RH. However, on the whole, the ambient condition of low temperature and high RH favored $NO_3^-$ formation, which is consistent with a previous report [37]. To summarize, the concentrations of $SO_4^{2-}$ and $NO_3^-$ depended not only on the contents of their precursors such as $SO_2$ and $NO_2$, but also on environmental conditions such as atmospheric oxidizing capacity, T and RH.

The mass ratio of $NO_3^-/SO_4^{2-}$ has been usually regarded as an indicator of the relative importance of mobile and stationary sources of sulfur and nitrogen in the atmosphere [37,61]. High $NO_3^-/SO_4^{2-}$ mass ratio means the predominance of mobile source over stationary source of pollutants [61]. The average value of $NO_3^-/SO_4^{2-}$ was 1.17 ± 0.71 in the six double high pollution cases. This result was close to that reported by Kong et al. (i.e., 1.00 during haze days in 2011) [48], but it also showed a marked difference from the other observations in Shanghai, for example, 0.67 in 2009 and 0.86 in 2012 [62]. Furthermore, the mass ratio value of our study was also higher than that of other megacities in China where average mass ratios were 1.02 in 2012 and 0.1–0.3 in 2006 in Beijing and Guangzhou, respectively [63]. The higher value in this study indicated that mobile vehicle emissions were making a significant contribution to $PM_{2.5}$ in Shanghai. This is consistent with the large vehicle population in Shanghai. In addition, Shanghai is not only located at the intersection of the East China Sea, but also on the waterway of the Yangtze River and Huangpu River, which included many mobile sources such as ship emissions. Therefore, the result showed the vital role of mobile sources in the double high pollution episodes in Shanghai. Besides industrial emissions, perhaps these sources emitted a lot of $O_3$ precursor $NO_2$, making Shanghai urban area be a VOC-limited regime of $O_3$ formation.

### 3.2.3. Characteristic of Gaseous Pollutants and Meteorological Parameters during the Double High Pollution Episodes

Secondary sulfate and nitrate aerosols in the atmosphere are primarily originated from the conversions of $SO_2$ and $NO_X$ released by anthropogenic sources. As shown in Figure 4b, the concentrations of $SO_2$ and $NO_2$ were affected by the morning and evening traffic rushes and the active photochemical processes. The enhancements of $NO_2$ concentrations were higher than those of $SO_2$ in most periods. The average mass concentrations of $SO_2$ and $NO_2$ during the whole double high pollution cases were 22.15 ± 13.70 and 39.04 ± 17.40 μg/m$^3$, respectively (Table 1). The average mass concentration of $SO_2$ was about 1.39 times higher than that in the double low periods (17.11 ± 8.67 μg/m$^3$). However, the average $NO_2$ concentration in the double high cases was close to that in the double low cases (38.67 ± 21.96 μg/m$^3$). For their specific changes, $SO_2$ was taken as an example. The concentration of $SO_2$ increased starting from 6:00 a.m. and reached a peak value at 8:00–9:00 a.m., and after 9:00 a.m., $SO_2$ concentration generally decreased until 14:00, and then it reached a secondary higher value of one day, which coincided with $SO_4^{2-}$ formation. The rapid formation of sulfate during this period could be mainly attributed to $SO_2$

oxidation under active photochemical processes. The appearance of the secondary higher value is earlier than evening rush hours, suggesting that $SO_2$ was mainly emitted from stationary emissions rather than mobile emissions. From 9:00–16:00 the mass concentration of $SO_4^{2-}$ increased continuously, suggesting that photochemical processes induced by solar radiation still played a significant role in $SO_2$ oxidation into $SO_4^{2-}$, though other oxidizing species such as $NO_2$ and $O_3$ may also play a role in $SO_2$ oxidation. After 16:00, both $SO_2$ and $SO_4^{2-}$ showed decreasing trends. Noticeably, $SO_2$ concentration after 20:00 increased slightly, while no increasing trend of $SO_4^{2-}$ was found, which confirmed the importance of sunlight in the formation processes of secondary sulfate aerosols.

Meanwhile, meteorological factors also played a key role in the formation processes of these secondary pollutants. The average RH and WS in the double high pollution cases were 63.6% $\pm$ 17.4% and 1.9 $\pm$ 1.2 m/s (Table 1), and the wind speed in Figure 4e showed an overall calm condition. Wind speed less than 3 m/s could provide a stagnant atmospheric condition [64], which played a critical role in the formation and accumulation of $O_3$ and secondary species. The moderately high RH condition could enhance the hygroscopic growth of aerosol to form liquid droplets or water films on aerosol surfaces, and hence promote the heterogeneous processes for SIA formation [4,48]. As a result, moderately high RH and low WS together promoted the formation and accumulation of SIA and then further led to high concentration of $PM_{2.5}$.

Furthermore, Figure 5 showed the diurnal variations of surface boundary layer height (BLH) and solar radiation (SR) under the six double high pollution cases. Considering high $PM_{2.5}$ concentrations during the double high pollution periods, the SR could be lower than that during the double low periods due to light scattering of the high-level $PM_{2.5}$ [65]. Therefore, the high loading of fine particles would reduce $O_3$ concentration owing to its dimming effect [16]. However, $O_3$ concentration increased rapidly and immediately after $PM_{2.5}$ and then presented an overall simply unimodal at 13:00–15:00 throughout the whole day in the double high pollution cases. This result implied the photochemical formation of $O_3$ to some extent, though low solar radiation during the double high pollution cases rendered strong photochemical activity impossible. Combined with the changes of $O_3$ and $PM_{2.5}$ concentrations in Figure 4a, the daily variation trends of SR were consistent with those of $O_3$ and $PM_{2.5}$. This clearly indicated that there still existed active photochemical processes in the six double high pollution periods, which not only favored the formation of $O_3$ in the presence of high concentration $PM_{2.5}$, but also favored the enhancement of atmospheric oxidizing capacity and further promoted the formation of secondary aerosols and the increase of $PM_{2.5}$ concentrations. This was consistent with our previous discussion. Furthermore, as shown in Figure 5, BLH was always at a high level during the high solar radiation period of each day, and showed a change characteristic of first increasing and then decreasing, which was not only conducive to the enhanced formation of pollutants such as $O_3$ and $PM_{2.5}$ in the increasing period of BLH, but also conducive to the subsequent increase of $O_3$ and $PM_{2.5}$ concentrations in the decreasing period of BLH, promoting the formation and accumulation of secondary particulate pollutants in the atmospheric boundary level and resulting in serious mixed pollution episodes with high-levels of $PM_{2.5}$ and $O_3$.

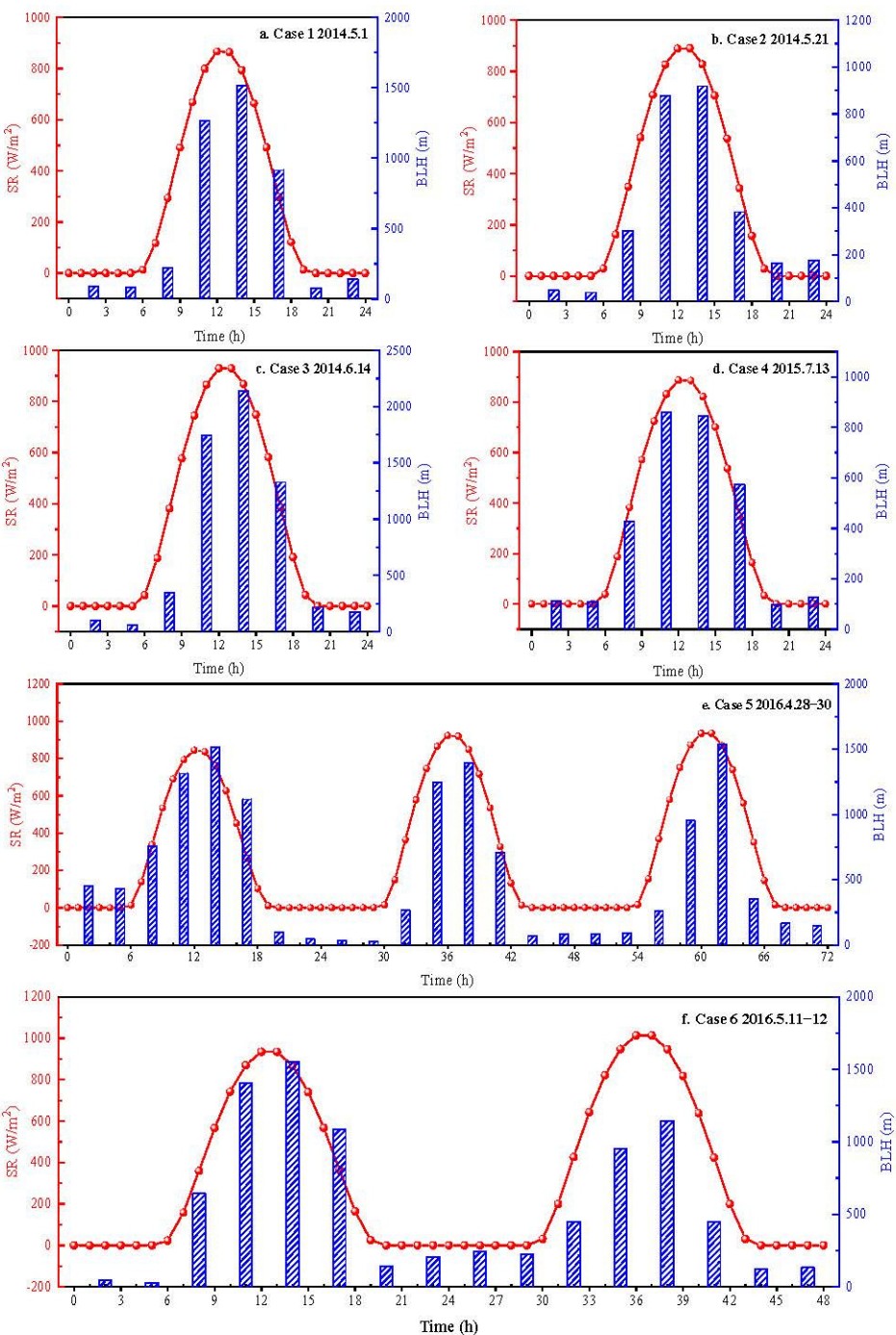

**Figure 5.** Variations of boundary layer height (BLH) and global solar radiation (SR) at ground level during the double high-level $O_3$ and $PM_{2.5}$ pollution cases. (**a**) Case 1; (**b**) Case 2; (**c**) Case 3; (**d**) Case 4; (**e**) Case 5; (**f**) Case 6.

### 3.3. Formation Mechanisms of the Double High-Level $O_3$ and $PM_{2.5}$ Episodes

To further understand the formation mechanisms of the double high-level $O_3$ and $PM_{2.5}$ episodes, except for the correlations mentioned above, the correlations of $O_3$, $PM_{2.5}$, $NO_2$, NO, T, RH, and visibility (Vis) observed in each mixed pollution case were analyzed (Table S3). The correlations of $PM_{2.5}$ with $O_3$ were positive in Case 2 (0.869), Case 3 (0.621), and Case 4 (0.828), and they were significant at the 0.01 level (two-tailed). These results not only suggested that a high concentration of $O_3$ could coexist with a high concentration of $PM_{2.5}$, but also implied that a high concentration of $O_3$ increased atmospheric oxidation, promoted the formation of secondary aerosols, and further increased the concentration

of $PM_{2.5}$. A strong positive correlation of temperature with $O_3$ was found in all the pollution cases, and the highest was 0.925 in Case 4. This result indicated that temperature could significantly affect the generation of $O_3$, and this finding is similar to previous studies [22]. In contrast, RH presented a strong negative correlation with $O_3$ concentration. The highest correlation value was $-0.919$ in Case 4 (RH $\approx$ 65%). This result suggested that water molecules may destroy $O_3$ or the water film on particle surface or a droplet in the atmosphere may dissolve $O_3$, and thus affect its production and accumulation, and therefore too high RH is not conducive to the formation and accumulation of gaseous $O_3$, while moderate RH is conducive to the existence of high concentration of $O_3$. In general, $O_3$ concentration has a strong positive correlation with temperature and solar radiation in summer [66], when it is usually with high visibility; meanwhile, under these circumstances, $PM_{2.5}$ concentration was often at a moderately low level. However, in this study, the correlations of $PM_{2.5}$ and visibility were negative in Cases 4–6 (r = $-0.800$, $-0.812$, and $-0.808$, respectively), and no correlation was found in Cases 1–3 (r = $-0.256$, 0.202, and 0.113, respectively). This result further suggested that $PM_{2.5}$ and $O_3$ could significantly affect the generation of each other. The negative correlations of $PM_{2.5}$ and RH were derived in all the pollution cases except Case 5 and 6, which may be due to the complex formation mechanism of secondary aerosols in $PM_{2.5}$ under different RH. For example, secondary sulfate aerosols in the atmosphere are usually formed through gas-phase oxidation at low RH, heterogeneous oxidation at moderate RH, and aqueous-phase oxidation at high RH [67]. The variations of RH in the six pollution episodes obviously increased the complexity of sulfate aerosol formation, and interfered with the correlation between $PM_{2.5}$ and RH. Meanwhile, the positive correlations between $PM_{2.5}$ and temperature were illustrated in all cases except Cases 5 and 6. Combining the results of $PM_{2.5}$ and $O_3$ with T and RH, we found that the conditions of high T and moderate RH was generally conducive to co-pollution of $O_3$ and $PM_{2.5}$. As for the important precursor of $O_3$ formation in photochemical reactions, $NO_2$ has moderate or weak negative correlations with $O_3$ in all cases. This may be due to the complex sources of $NO_2$ and the formation of some $NO_2$ via the oxidation of NO by $O_3$. Negative correlations between $O_3$ and NO were found in all cases, which was mainly attributed to the titration effect of NO, which confirmed the oxidation reaction of NO by $O_3$.

### 3.4. Impact of Daytime HONO on $O_3$ Formation during Double High-Level $PM_{2.5}$ and $O_3$ Pollution Cases

MARGA used pure water in WRD to absorb some trace gases, including HONO. The HONO measured by MARGA may be affected by the hydrolytic disproportionation of atmospheric $NO_2$. However, $NO_2$ (g) is weakly dissolved in pure water (Henry's law constant H $\sim$ 0.01 M atm$^{-1}$), and its uptake coefficient on pure water is very small ($\gamma \sim 1 \times 10^{-7}$) [68]. Therefore, the formation of HONO from the dissolution of $NO_2$ (g) in pure water is unfavorable for both kinetic and thermodynamic reasons, and the influence of atmospheric $NO_2$ hydrolytic disproportionation on the measurement of HONO can be ignored. HONO data can be used to investigate the role of HONO in the formation of $O_3$ during the double high-level $PM_{2.5}$ and $O_3$ pollution episodes.

As mentioned in the introduction, several studies have reported that HONO photolysis was the dominant OH source. For example, HONO photolysis contributed 80.4% of atmospheric primary OH production in the wintertime [27], or more than 92% of atmospheric primary OH production in polluted areas [28]. Such a large contribution should greatly affect the formation of hydroperoxyl and organic peroxy radicals in the atmosphere, and then further significantly affect the formation of $O_3$, and hence, the dominant contribution of HONO photolysis to primary OH production may increase the linear dependence between HONO and $O_3$. Therefore, the relationships of daytime HONO and $NO_2$ with $O_3$ in each double high pollution case were analyzed to reveal the possible role of daytime HONO in $O_3$ formation.

Table 2 showed the correlation between $O_3$ and HONO (or $NO_2$) in each double high pollution case during the daytime. As shown in Table 2, negative correlations between

HONO (or $NO_2$) with $O_3$ were found during the daytime in all cases, indicating their consumption during the formation of $O_3$. For the relationship of HONO with $O_3$, there existed a strong negative correlation during 6:00–18:00 (r = −0.860, −0.917, −0.918, −0.688, −0.811, and −0.769 for Cases 1–6, respectively). However, during the enhanced solar radiation period (8:00–16:00), the strong linear dependence between HONO and $O_3$ still existed in Cases 1–3 (r = −0.871, −0.916, and −0.935, respectively), Case 5 (r = −0.900), and Case 6 (r = −0.676), revealing the dominant role of HONO in $O_3$ formation during these pollution episodes. Meanwhile, this result may in turn suggest that HONO photolysis made a great contribution to atmospheric primary OH production during these pollution episodes. While the linear dependence between HONO and $O_3$ during the enhanced solar radiation periods (8:00–16:00) was weakened in Case 4 (r = −0.385). Case 4 had the highest average T (30.5 ± 3.0 °C), the lowest average WS (1.1 ± 0.3 m/s), and the highest average HONO (3.86 ± 0.99 μg/m$^3$) and $NH_3$ (18.11 ± 1.03 μg/m$^3$) concentrations, as well as the lower average $NO_3^-$ concentration (9.62 ± 4.47 μg/m$^3$) during 8:00–16:00 among the six cases (Table S4). The high temperature was not conducive to the stability of ammonium nitrate, because the formed ammonium nitrate was thermodynamically unstable, which may elevate $NH_3$ concentration and decrease $NO_3^-$ concentration. Moreover, the observation of high HONO concentration at high temperature in Case 4 may imply the impacts of OH from other sources on the $O_3$ formation or the changes of HONO sources and $O_3$ formation pathways under the enhanced solar radiation. For example, OH radicals can also be produced by the photolysis of atmospheric nitrate aerosol, nitric acid, and $H_2O_2$ under solar radiation in the atmosphere [69,70], and the photolysis of atmospheric nitrate aerosol can also produce $O(^3P)$ [70], which may prompt the formation and accumulation of $O_3$. Additionally, the high average $NH_3$ concentration in Case 4 during the daytime may induce explosive growth in HONO by $NH_3$-promoted hydrolysis of $NO_2$ [71], which emerged as a relatively prominent HONO source compared to the other cases. All the mentioned above would disturb the linear dependence between daytime HONO and $O_3$ in Case 4. Meanwhile, it should be pointed out that the enhanced solar radiation in Case 4 could promote the photochemistry of $NO_2$ and thus enhance the negative correlation between $NO_2$ and $O_3$ (Table 2). In addition, compared to those in the double low cases (Table S5), there existed overall a stronger linear dependence between HONO and $O_3$ during the daytime (6:00–18:00) in all the cases except Case 4, revealing a more dominant role of HONO in $O_3$ formation and a more active atmospheric photochemical process during the double high pollution episodes. In short, the daytime HONO concentration was, on the whole, strongly negatively correlated with $O_3$ concentration during the double high pollution episodes, which not only revealed the dominant contribution of HONO photolysis to atmospheric primary OH production, but also indicated the indispensable role of HONO in $O_3$ formation during the double high pollution episodes.

**Table 2.** Correlation between $O_3$ and HONO (or $NO_2$) in each double high pollution case during daytime.

| Double High Case | | HONO | | NO₂ | |
|---|---|---|---|---|---|
| | | 6:0–18:00 | 8:0–16:00 | 6:0–18:00 | 8:0–16:00 |
| Case 1 | $O_3$ | −0.860 ** | −0.871 ** | −0.822 ** | −0.915 ** |
| Case 2 | $O_3$ | −0.917 ** | −0.916 ** | −0.658 ** | −0.962 ** |
| Case 3 | $O_3$ | −0.918 ** | −0.935 ** | −0.825 ** | −0.898 ** |
| Case 4 | $O_3$ | −0.688 ** | −0.385 | −0.208 | −0.525 |
| Case 5 | $O_3$ | −0.811 ** | −0.900 ** | −0.683 ** | −0.707 ** |
| Case 6 | $O_3$ | −0.769 ** | −0.676 ** | −0.602 ** | −0.426 |

** with significant value at *p* < 0.01.

Compared to those in the double low cases during the daytime from 6:00 to 18:00 (Table S6), the average concentrations of HONO, $O_3$, $SO_4^{2-}$, $NO_3^-$, SIA, and $PM_{2.5}$ on the double high pollution days were 1.18, 2.57, 2.62, 1.44, 1.90, and 2.12 times higher than those of the double low days (Table S4). The elevated $O_3$ on the double high pollution

days suggested the enhanced formation of $O_3$ and the enhanced atmospheric oxidizing capacity during the double high pollution cases, which would promote conversions of $SO_2$ and $NO_2$, and hence the enhanced formation of secondary $SO_4^{2-}$, $NO_3^-$, SIA, and $PM_{2.5}$ was observed. However, it should be pointed out that the variable positive or negative correlations between daytime HONO and aerosol species (i.e., $SO_4^{2-}$, $NO_3^-$, SIA, and $PM_{2.5}$, Table S7) showed that the conversions of $SO_2$ and $NO_2$ were not entirely oxidized by HONO and its photolysis product OH, but they were likely to be oxidized by the other oxidizing species produced during the photochemical processes initiated by OH. Therefore, this result indicated the direct and indirect impacts of HONO on the formation of secondary species. Finally, $K^+$ ion is the most commonly used tracer to judge and identify biomass burning pollution source [72], while biomass burning in agricultural activities can cause high $PM_{2.5}$ and $O_3$ pollution in the YRD in China [17]. However, the low $K^+$ concentration ($0.88 \pm 0.54$ μg/m$^3$, concentration range: $0.47 \pm 0.17 - 1.58 \pm 0.47$ μg/m$^3$) during the daytime in the double high pollution episodes excluded the impact of biomass burning events on $O_3$ formation [73].

## 4. Conclusions

This study focused on investigating the pollution characteristics and occurrence mechanisms of double high-level $O_3$ and $PM_{2.5}$ pollution episodes and the significant role of daytime HONO in $O_3$ formation in these pollution episodes, based on almost-three-year observation measurements at the Pudong New Area environmental monitoring station in Shanghai. Results showed that high fine particulate pollution and high $O_3$ pollution could occur simultaneously. Even if the frequency of the occurrence of the double high pollution accounted for 1.0% in the whole observation period, the formation of $PM_{2.5}$ and $O_3$ was intertwined and promoted mutually, which led to synchronous pollution both on long time scales and large regional scales and caused more serious air pollution. During the double high-level $O_3$ and $PM_{2.5}$ pollution episodes there still existed active photochemical processes, while the active photochemical processes at high $PM_{2.5}$ concentration favored the production and accumulation of $O_3$ under a calm atmospheric condition, including relatively high T, moderate RH, and low WS, which in turn enhanced the conversions of $SO_2$ and $NO_2$ and the formation and accumulation of secondary sulfate and nitrate aerosols, and further increased the concentration of $PM_{2.5}$ and promoted the evolution of air pollution.

Furthermore, the daytime HONO concentration was, on the whole, strongly negatively correlated with $O_3$ concentration during the active photochemical processes of the double high pollution episodes, indicating that HONO plays a dominant role in the formation of $O_3$ in the double high pollution episodes. This result not only reveals the dominant contribution of HONO photolysis to the production of atmospheric OH and the indispensable role of HONO in $O_3$ formation during the double high pollution episodes, but also indicates the enhanced atmospheric oxidizing capacity by producing various oxidizing species for the oxidation of $SO_2$ and $NO_2$ during the double high pollution cases, which would promote the formation of SIA and further elevate the concentration of $PM_{2.5}$.

This study not only further confirmed that high fine particulate pollution and high $O_3$ pollution could occur simultaneously under suitable atmospheric conditions, but also highlighted the indispensable role of HONO in $O_3$ formation during the double high-level $PM_{2.5}$ and $O_3$ pollution episodes, though HONO sources are still not well understood up to now, and therefore, it is very important to understand the additional HONO sources in the polluted air while collaborative control measures of $O_3$ and $PM_{2.5}$ precursors have been carried out to improve the air quality in Shanghai and YRD region.

**Supplementary Materials:** The following are available online at https://www.mdpi.com/article/10.3390/atmos12050557/s1, Figure S1: Wind speed and directions at the observation site in December of 2014–2016, Figure S2: Variations of SOR, NOR, $SO_2$, $NO_2$, $SO_4^{2-}$, $NO_3^-$ in different double high-level $O_3$ and $PM_{2.5}$ pollution cases, Table S1: Statistics of meteorological parameters during observation period 2014–2016, Table S2: Summary of meteorological factors and chemical species data in the

double low-level O$_3$ and PM$_{2.5}$ pollution cases, Table S3: The correlation of ambient air pollutants and meteorological parameters in each double high-level pollution case, Table S4: Summary of meteorological factors and chemical species data in the double high-level O$_3$ and PM$_{2.5}$ pollution cases during the daytime, Table S5: The correlation between O$_3$ and HONO (or NO$_2$) in each double low case during daytime, Table S6: Summary of the concentration data of HONO, O$_3$ and aerosol species in each double low case during the daytime, Table S7: The correlations between the daytime HONO and aerosol species (i.e., SO$_4^{2-}$, NO$_3^-$, SIA and PM$_{2.5}$) during the double high episodes.

**Author Contributions:** Conceptualization, L.K.; Investigation, K.Y., L.K. and F.S.; Validation, K.Y., L.K., J.S. and F.S.; Data curation, K.Y., S.J., L.K. and F.S.; Formal analysis, K.Y. and L.K.; Visualization, K.Y., S.T., L.C., C.W. and L.K.; Writing–original draft preparation, K.Y.; Writing–review and editing, K.Y., S.T., L.C., C.W., S.J., L.K. and L.W.; Funding acquisition, L.K. and L.W.; Project administration, L.K.; Supervision, L.K. All authors have read and agreed to the published version of the manuscript.

**Funding:** This research was funded by the National Natural Science Foundation of China (Grant Nos. 21777027, 21976032, and 41475110) and the National Key R & D Program of China (2017YFC0209505).

**Institutional Review Board Statement:** Not applicable.

**Informed Consent Statement:** Not applicable.

**Data Availability Statement:** The data reported in this study will be available on request from the corresponding author.

**Conflicts of Interest:** The authors declare that they have no conflict of interest.

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
