# Peer review of "Double High-Level Ozone and PM2.5 Co-Pollution Episodes in Shanghai, China: Pollution Characteristics and Significant Role of Daytime HONO"

_atmosphere, doi:10.3390/atmos12050557_

Round 1

Reviewer 1 Report

  1. Change O3 in abstact and Introduction to O3.
  2. Put  the header in Table 1 on the next side as well.
  3. The font in Table 2 is difficult to read.
  4.  Acording to your results is there a need to  revise the current  air quality guidelines for PM2.5, ozone or other pollutants in your country? Put your opinion in Conclusion.

Reviewer 2 Report

The paper “Double high-level ozone and PM2.5 co-pollution episodes in 2 

Shanghai, China:pollution characteristics and significant role 3 of daytime HONO” reports on 

the role of HONO in O3 formation during the daytime in episodes of double high level PM25 and O3  pollution,  investigated based on field observation in urban Shanghai from 2014 to 2016 

This is in general an interesting topic.

This paper present a consistent and interesting dataset with a lot of potentiality. 

Unfortunately the results are not well presented and the manuscript lack in focus which hamper 

any robust conclusion.

My impression is that the text is too long with many details that are not directly relevant to the paper main objectives. The paper reading, results and conclusions will greatly benefit from a substantial reduction, deleting all the descriptive and relatively well know parts.

Furthermore an additional analysis of long range transport of O3 is needed to exclude this potential mechanism for the high level recorded in specific episodes.

Main points

- The TEOM measured mass has a strong temperature and RH dependence (underestimation). The effect is larger at larger concentrations. Loss of semi volatiles compounds is the probable cause.  This should be introduced and discussed. 

See I.e. 

Allen et al. 

JOURNAL OF THE AIR & WASTE MANAGEMENT ASSOCIATION Volume: 47 Issue: 6 Pages: 682-689

DOI: 10.1080/10473289.1997.10463923

AND

Charron et al. 

ATMOSPHERIC ENVIRONMENT

Volume: 38 Issue: 3 Pages: 415-423

DOI: 10.1016/j.atmosenv.2003.09.072

- Double high episodes:

Line 360-362: the conditions are defined here. Some statistics will greatly help to understand the relevance of the present work: how many of these episodes have been recorded during the three years period, a part the 6 case studies, described in detail? What about seasonal dependences?

In all the episodes O3 seems to be clearly high but PM25 not so clearly higher than the average. Could the author add some statistics on this point?

-line 450-460: SOR and NOR are higher during double high episodes with respect to previous studies in Shanghai. SOR and NOR for low cases are also higher than those observed in previous studies. Please comment. 

-how the authors can exclude that O3 is transported from medium or long range to the receptor site, during the case studio episodes?

- the conclusion (lines 749-757) on the role of HONO on O3 formation are vague. Is it HONO relevant or not? 

Style 

1.in many places, especially in the beginning subscripts are missing, for example PM25 or SO4=…please correct everywhere (I.e. lines 45, 51,….)

2. Too many digits for PM25 and O3 concentrations. Fort example line 333 “43.48 pm 25.38” should be “43.5 pm 25” and so on….(in table 2 also) 

As regards the text structure many part can be shortened or deleted.

For example:  

-abstract lines 14-20 (rephrase + reduce) 

-Table 1 (move to SM)

- 3.1.1 Seasonal behaviors (this must be greatly reduced focussing only on PM25 and O3). Please add some statistics on PM25 (numbers rather than qualitative discussion) . 

-lines 337-339

-lines 352-356

-lines 728-733 

and many others places…

Finally, a substantial rewriting of the discussion is necessary to focus on the real results.

Minor points

Line 325. Delete “most”

Reviewer 3 Report

The present research evaluates the ozone and PM2.5 co-pollution episodes in Shanghai. First of all, I consider this topic of high importance and interest to the scientific community. In fact, I agree that the introduction correctly exhibits the value of this research and the authors suggest a good hypothesis. In general, the article has good content but important errors in format and long sentences have been detected (see attach).

However, the main lack of this article is the absence of statistical. At the end of the article, the authors evaluated a possible correlation between O3 and HONO (or NO2) in different cases. But during the rest of the article, they are only observed the data obtained and carried out hypotheses based on the observation (when they have the data available). For example, they have all the climate conditions data and the O3, PM2.5., etc. information of each month. They only make correlations based on linear models with bad coefficients (r < 0.8 in most of the cases), instead of carried out an ANOVA or other statistical analysis that could clarify a possible correlation.

For all these reasons, I consider that this article should be enhanced before being published in Atmosphere journal.

Round 2

Reviewer 2 Report

The paper has been improved by the authors and I think it is now close to be publishable. The only last, but  important, revision I'm asking is to remark, both in the abstract and in the conclusions, the statistical significance of the double high level episodes, ~1%, on the total days in a three year period. This is relevant to provide a flavour of the environmental significance of these episodes. 

Author Response

Thank you for you careful suggestion.  We have added the statistical and environmental significance in the revised manuscript. â‘  Line 23-24 in Abstract (highlighted in yellow-colored text) â‘¡Line 700-703 in Conclusion (highlight in yellow-colored text) â‘¢ Other language mistakes have been corrected in the revised manuscript (marked with yellow background in the context).

Reviewer 3 Report

I consider that the modifications carried out by the authors enhancing the article quality and I recommend this manuscript to be published.

Author Response

Thank you for your comments. We have checked the language and spell again in the revised manuscript. The revised parts have been marked with yellow background in context.